# Functional Proteomic Profiling Analysis in Four Major Types of Gastrointestinal Cancers

**DOI:** 10.3390/biom13040701

**Published:** 2023-04-20

**Authors:** Yangyang Wang, Xiaoguang Gao, Jihan Wang

**Affiliations:** 1School of Electronics and Information, Northwestern Polytechnical University, Xi’an 710129, China; 2Institute of Medical Research, Northwestern Polytechnical University, Xi’an 710072, China

**Keywords:** gastrointestinal cancer, TCPA, feature selection

## Abstract

Gastrointestinal (GI) cancer accounts for one in four cancer cases and one in three cancer-related deaths globally. A deeper understanding of cancer development mechanisms can be applied to cancer medicine. Comprehensive sequencing applications have revealed the genomic landscapes of the common types of human cancer, and proteomics technology has identified protein targets and signalling pathways related to cancer growth and progression. This study aimed to explore the functional proteomic profiles of four major types of GI tract cancer based on The Cancer Proteome Atlas (TCPA). We provided an overview of functional proteomic heterogeneity by performing several approaches, including principal component analysis (PCA), partial least squares discriminant analysis (PLS-DA), t-stochastic neighbour embedding (t-SNE) analysis, and hierarchical clustering analysis in oesophageal carcinoma (ESCA), stomach adenocarcinoma (STAD), colon adenocarcinoma (COAD), and rectum adenocarcinoma (READ) tumours, to gain a system-wide understanding of the four types of GI cancer. The feature selection approach, mutual information feature selection (MIFS) method, was conducted to screen candidate protein signature subsets to better distinguish different cancer types. The potential clinical implications of candidate proteins in terms of tumour progression and prognosis were also evaluated based on TCPA and The Cancer Genome Atlas (TCGA) databases. The results suggested that functional proteomic profiling can identify different patterns among the four types of GI cancers and provide candidate proteins for clinical diagnosis and prognosis evaluation. We also highlighted the application of feature selection approaches in high-dimensional biological data analysis. Overall, this study could improve the understanding of the complexity of cancer phenotypes and genotypes and thus be applied to cancer medicine.

## 1. Introduction

Gastrointestinal (GI) cancer refers to malignancies of the GI tract and digestive organs. GI cancer accounts for one in four cancer cases and one in three cancer-related deaths globally (Source: Cancer Today, https://gco.iarc.fr (accessed on 23 January 2023)). Several major types of GI cancers, such as oesophagus (approximately 570,000 new cases in 2018), stomach (1.0 million cases), colorectum (1.8 million cases), liver (840,000 cases), and pancreas cancer (460,000 cases), are largely distinct with respect to aetiologies, epidemiologic distributions [1], environmental risk factors, prevention strategies, and lifestyles. A deeper exploration of the complexity of cancer phenotypes and genotypes will improve the understanding regarding cancer development and malignant progression mechanisms, and this knowledge can be further applied to cancer medicine [2].

The diversity of cancer covers many factors, including genetics, cell/tissue biology, pathology, response to therapy, and more [2]. Over the past few decades, comprehensive sequencing applications have revealed the genomic landscapes of the common types of human cancer [3]. These studies have demonstrated that intragenic mutations of “drive genes” can prompt or “drive” tumorigenesis [3]. Mechanistically, casual or inherited mutations of critical genes can regulate cell growth and differentiation and encode DNA repair proteins, which induce malignancy oncogenesis and progression. In addition, transcriptional changes or differentially expressed genes (DEGs) contribute to cancer initiation and metastatic progression [4]. Studies have presented a novel bioinformatics pipeline that could distinguish tumour from normal tissues based on DEGs across 10,704 tumour and normal samples from The Cancer Genome Atlas (TCGA) [5]. For GI cancers, studies have shown that gastric cancer (GC) transcriptome analysis helps in identifying histotype-specific molecular signatures with prognostic potential [6]. Screening differentially expressed immune-related genes (IRGs) in colon adenocarcinoma (COAD) that will benefit cancer immunotherapy and immunomodulation [7].

With more successes in sequencing genomes, an emerging frontier is the proteome, that is identifying and studying expressed proteins in the human body and other organisms [8]. Proteomics has developed as a crucial tool for exploring biological changes in cancer. Important information, including protein targets and signalling pathways related to cancer growth and progression, has been identified through proteomics technology [9]. The Cancer Proteome Atlas (TCPA) provides a comprehensive bioinformatic resource for assessing, visualizing, and analysing the functional proteomics data of two separate applications, including patient tumour and cell line samples (http://tcpaportal.org (accessed on 2 February 2023)) [10,11]. The first part focuses on reverse-phase protein array (RPPA) data of patient tumours, containing more than 8000 samples across 32 cancer types from TCGA and other independent patient cohorts, which provides a great resource for researchers who are analysing functional proteomics in different cancer types.

Therefore, the aim of this study was to comprehensively explore the functional proteomic profiles of four major types of GI tract cancer based on the TCPA and TCGA databases, including oesophageal carcinoma (ESCA), stomach adenocarcinoma (STAD), COAD, and rectum adenocarcinoma (READ). Here, we provide an overview of the functional proteomic heterogeneity in the four types of GI tumours. We further applied feature selection approaches (detailed information is described in Materials and Methods) during the data analysis to screen candidate protein signature subsets to better distinguish different cancer types. Feature selection methods present the merit of acquiring more informative and compact molecular features than those obtained by traditional means and thus play an important role in machine learning-based classification tasks, especially in high-dimensional data, such as biological omics datasets [12]. In recent decades, a large variety of feature selection methods have been widely developed and utilized in medicine and biology fields, which can be used to identify the critical genome/proteome signatures in the corresponding expression dataset with thousands of dimensions [13,14,15,16]. Filter-based feature selection is more popular than ever since these methods are more suitable for high-dimensional datasets with less computational complexity and can rank the features without the need for training classifiers. Finally, the potential clinical implications of candidate proteins in terms of tumour progression and prognosis were also evaluated in this study.

## 2. Materials and Methods

### 2.1. Subsection Acquisition and Preprocessing of TCPA and TCGA Datasets

We obtained RPPA functional proteome profiles of ESCA, STAD, COAD, and READ from the TCPA portal (https://tcpaportal.org/tcpa/ (accessed on 5 February 2023)). According to the guidelines, when analysing the RPPA data, the merged Pan-Can L4 data should be used for multiple disease analysis. Thus, the whole original dataset of the “TCGA-PANCAN32-L4.zip” file was downloaded from TCPA. The RPPA proteome dataset consists of the relative abundances of 258 protein markers in tumour samples. We discovered that 41 protein markers had missing values (“NA”) in more than half (51.92%~90.86%) of the total samples and were then deleted. Then, RPPA proteome profiling including 217 proteins in tumour samples of ESCA, STAD, COAD, and READ was extracted from the whole dataset for further analysis in this study.

The clinical phenotype and survival information corresponding to the tumour samples of ESCA, STAD, COAD, and READ were obtained from the TCGA portal (https://tcga-data.nci.nih.gov/tcga/ (accessed on 5 February 2023)). We combined the three documents, including tumour RPPA proteome profiling, phenotype information, and survival information, through sample ID for further analysis.

### 2.2. Functional Proteome Profiling Analysis in the Four Types of Gastrointestinal Cancers

To gain a system-wide understanding of the four types of gastrointestinal cancer on the basis of RPPA functional proteome profiling, we performed several approaches, including principal component analysis (PCA), partial least squares discriminant analysis (PLS-DA), t-stochastic neighbour embedding (t-SNE) analysis, and heatmap analysis, to obtain a basic overview of the tumour sample distributions. Specifically, the PCA, PLS-DA, t-SNE, and heatmap were conducted with the “PCA” (in “FactoMineR” package), “plsda” (in “mixOmics” package), “Rtsne” (in “Rtsne” package), and “pheatmap” (in “pheatmap” package) algorithms in R 4.0.2, respectively. The source code for the clustering methods is available on https://github.com/jihanwang/FourClusterMethods (accessed on 7 February 2023).

### 2.3. Using Feature Selection Approaches to Identify Protein Signatures for Classifying Different Cancer Types

Feature selection is used to obtain an optimal subset from original features for model building. As a basic tool derived from information theory, mutual information (MI) is a measure for two random vectors, and different mutual information based on feature selection methods has been proposed. The mutual information between random variables of $X={\left(x_{1},x_{2},\ldots ,x_{m}\right)}^{T}$ and $Y={\left(y_{1},y_{2},\ldots ,y_{m}\right)}^{T}$ is defined as:(1)$$I(X;Y)=\sum_{x_{i}\in X}\sum_{y_{j}\in Y}p(x_{i},y_{j})log\frac{p(x_{i},y_{j})}{p(x_{i})p(y_{j})}$$

The max-relevance and min-redundancy (mRMR) feature selection framework is a criterion that considers not only the relevance between feature $f_{k}$ and target C but also the redundancy as a penalty for removing similar features. The criterion of mRMR is as follows:(2)$$J(f_{k})=\arg\max_{f_{k}\in F-S}(I(f_{k};C)-\frac{1}{\left|S\right|}\sum_{f_{j}\in S}I(f_{k};f_{j}))$$
where $J(f_{k})$ is the objective function, F is the original feature set, S is the selected feature subset, and $f_{k}$ is the candidate feature.

Although mRMR is convenient to rank the candidate features for discrete random variables, these datasets with continuous variables in medical research are more common and need to be discretized. To reduce the bias of discretization, K-nearest neighbour (KNN)-based MI estimation of the mutual information feature selection (MIFS) method can be used to obtain the MI between any two features without computation growing exponentially even for a large number of features. In this study, we used the combination of the KNN-based MI estimation method and mRMR for ranking the protein signatures and screened key biomarkers to build a model for classifying different cancer types. The implementation code was obtained from https://github.com/danielhomola/mifs (accessed on 10 February 2023).

### 2.4. Statistical Analysis

Comparisons of RPPA protein abundances among multiple cancer types were conducted by using one-way analysis of variance (ANOVA), with a *p* value < 0.05 indicating statistical significance. Correlation analysis between protein abundance and tumour stage (including stage I, II, III, IV) was performed with Spearman correlation analysis in R 4.0.2, with a *p* value < 0.05 representing a significant correlation. Univariate Cox regression analysis of overall survival (OS) was performed with the “coxph” algorithm (in the “survival” package) in R 4.0.2 to identify tumour prognosis-related factors. For Kaplan-Meier survival curve analysis, the candidate proteins were tested and visualized with the “survminer” package in R 4.0.2. The optimal cut-off value of protein abundance was determined by the “surv_cutpoint” function, using *p* < 0.05 as the test level in Kaplan-Meier analysis.

## 3. Results

### 3.1. Overview of the Functional Proteome Profiling across ESCA, STAD, COAD, and READ Tumour Samples

As described previously, after data preprocessing, we obtained RPPA functional proteome profiles including 217 protein markers in the samples of ESCA, STAD, COAD, and READ. We conducted PCA, PLS-DA, t-SNE, and heatmap clustering algorithms to explore the distribution and heterogeneity of tumour samples in accordance with the four types of GI cancer. As shown in Figure 1, the samples clustered significantly according to cancer type, which may indicate that different types of tumours possess relatively unique functional proteome profiles based on their tissue or origin. Moreover, we observed that tumour samples of COAD and READ overlapped significantly with each other in all four clustering models. Many studies have demonstrated that COAD shares similar molecular mechanisms with READ from multiomics perspectives [17,18]. Herein, we also verified the molecular similarity between COAD and READ on the basis of RPPA functional proteome profiling. As a result, we combined the two cancer types (COAD and READ) as one main cancer type of colorectal cancer (CRC) for further analysis in this study.

We then combined RPPA functional proteome profiling (derived from the TCPA portal) and phenotype characteristics as well as survival information (derived from the TCGA portal) corresponding to the tumour samples to explore the association between candidate proteins and clinical outcomes of GI cancers. Table 1 summarizes the clinical characteristics of ESCA, STAD, and CRC samples in this study.

### 3.2. Using Feature Selection Approaches to Identify Protein Signatures That Help to Classify Different Cancer Types

By performing feature selection algorithms, MIFS, we screened a subset of 20 protein signatures (including *MYH11*, *HER3_pY1289*, *CD20*, *STATHMIN*, *SMAD1*, *CHK1*, *P27_pT157*, *JAB1*, *PCADHERIN*, *IGF1R_pY1135Y1136*, *BCLXL*, *PREX1*, *PR*, *MIG6*, *ERCC1*, *CHK1_pS345*, *AR*, *CYCLIND1*, *HER3*, and *ADAR1*) for better classification among the ESCA, STAD, and CRC tumour samples. Detailed information and relative abundances of the 20 selected proteins are shown in Figure 2. ANOVA indicated significant differences in protein abundances among the ESCA, STAD, and CRC samples. By using the 20 selected protein markers, we observed better classifying models among the ESCA, STAD, and CRC samples, as shown in Figure 3. Specifically, the sum of dim 1 and dim 2 increased from 24.1% with all proteins (Figure 1A) to 50.5% (Figure 3A) with the 20 selected proteins in the PCA model. Similarly, in PLS-DA, the X-variate 1 and X-variate 2 explained 10% and 12% of the variability in the clusters using all proteins (Figure 1B),respectively, while the X-variate 1 and X-variate 2 explained 25% and 23% using the 20 candidate proteins (Figure 3B), respectively. Thus, the application of feature selection methods can improve the identification of marker/feature subsets of high-dimensional data in biomedical research.

### 3.3. Associations of Protein Biomarkers with the Clinical Characteristics of Tumours

Next, we investigated the associations of candidate protein biomarkers with tumour characteristics, mainly tumour stage and overall survival (OS) status, to explore the potential value of these proteins in tumour progression or prognosis. Spearman correlation analysis revealed that several proteins were associated with tumour stage in ESCA, STAD, and CRC tumours. As shown in Figure 4, the relative expression levels of *MYH11* and *CD20* were elevated in stages III/IV compared with stages I/II in both ESCA and STAD samples (Figure 4A,C) and were positively correlated with tumour stage (*p* < 0.05, Figure 4B,D). In CRC, the expression of *ADAR1* and *HER3* increased while *PREX1* levels decreased with stage progression from I to IV (Figure 4E); thus, we observed a positive correlation between *ADAR1* and *HER3* and a negative correlation between *PREX1* and the tumour stage of CRC (Figure 4F).

The survival analysis for identifying risk clinical parameters of tumour patients was conducted using the Cox proportional hazard model. In univariate Cox regression analysis, tumour stage was identified as a significant risk factor for poor overall survival in ESCA [hazard ratio (HR) = 2.2683, *p* < 0.05], STAD (HR = 1.5882, *p* < 0.01), and CRC (HR = 1.0242, *p* < 0.05) patients. In addition, higher age was associated with more prognostic risk in STAD (HR = 1.0296, *p* < 0.05) and CRC (HR = 1.0273, *p* < 0.05) patients, while lower BMI indicated better prognosis in CRC patients (HR = 0.9351, *p* < 0.05). Overall, other clinical parameters, including sex, race, and tumour neoplasm histologic grade, had no significant effects on overall survival in ESCA, STAD, and CRC patients (*p* > 0.05 in Cox regression) in our analysis, as shown in Figure 5.

Kaplan-Meier analysis was conducted to determine the correlation between protein expression and overall survival. The results in Figure 6 revealed that high expression of *CD20* and *PR* was associated with lower overall survival probability in ESCA and STAD patients; increasing levels of protein *AR*, *HER3*, *MYH11*, and *SMAD1* were also associated with poor overall survival in STAD patients. In CRC cases, elevated expression of *ADAR1*, *BCLXL*, *ERCC1*, *HER3*, and *PR* reflected a worse survival rate, whereas relatively high expression of *MYH11* showed a better survival rate. Taken together, these results suggested that some candidate proteins may be potential prognostic biomarkers for ESCA, STAD, and CRC patients.

## 4. Discussion

The present study characterized RPPA-based functional proteomic data in approximately 1000 tumour samples across four major types of GI tract cancer, including ESCA, STAD, COAD, and READ. The results revealed unique and common patterns in the four cancer cohorts, and the functional proteome signatures were relatively distinguishable in upper GI tract cancers, including ESCA and STAD, whereas the lower GI tract cancers of COAD and READ shared obviously similar functional proteome profiles in all clustering analyses (Figure 1). In our previous research, gut microbiome (GM) analysis also indicated relatively site/organ-specific microbial profiles across different GI cancer types [19]. However, similar to the current study, minor differences were observed in GM profiles between COAD and READ in the lower GI tract [19]. Thus, the COAD and READ cohorts are always considered two subgroups of the entire CRC cohort [20,21], which represents malignant conditions in the lower gastrointestinal tract.

In recent decades, more powerful experimental and computational tools/technologies have provided an avalanche of “big data” in cancer research [22]. Here, we highlighted the application of feature selection methods in cancer omics data analysis. Effective feature selection methods help to identify potential molecular biomarkers for further research and to train precise classifiers for different tumour type/subtype classifications or diagnoses [23]. Studies have demonstrated the application of feature selection in genomic analysis of STAD and COAD based on TCGA and Gene Expression Omnibus (GEO) cohorts [24,25]. In this study, we applied MIFS algorithms and screened the top 20 candidate protein markers in distinguishing ESCA, STAD, and CRC tumour samples. The relative abundances of the 20 selected proteins were significantly altered among ESCA, STAD, and CRC tumour samples according to ANOVA (Figure 2). In a study on prostate cancer, the texture features from transrectal ultrasound (TRUS) images were considered as variables and then ranked by the MIFS algorithm to classify cancerous and noncancerous tissues [26]. MIFS uses mutual information to measure the relevance between features and the target variable, which can capture both linear and nonlinear relationships between variables [27]. It is a powerful and flexible feature selection method that can help identify the most relevant features in a high-dimensional dataset, leading to better performance and more interpretable models, especially when handling missing data, noise, and outliers [28,29].

It is important to explore the biological significance of molecular biomarkers in the tumorigenesis, progression, or prognosis of cancers. Thus, we further investigated the associations of candidate proteins with tumour stage and overall survival status to evaluate the potential value of these proteins as progressive or prognostic markers. The results revealed that the expression levels of *CD20* and *MYH11* were positively correlated with the stages of ESCA and STAD, the two types of upper GI cancer (Figure 4A–D), and higher levels of *CD20* reflected a poorer overall survival rate in upper GI cancer (Figure 6A,B). As a B-cell surface marker, *CD20* is a transmembrane protein that is involved in B-cell development and differentiation [30]. *CD20* has been found to be expressed in several B-cell malignancies, such as chronic lymphocytic leukaemia, diffuse large B-cell lymphoma, mantle cell lymphoma, and follicular lymphoma [31,32,33], thus highlighting its therapeutic implications in B-cell malignancies [30]. *MYH11* is a contractile protein that functions in converting chemical energy into mechanical energy through adenosine triphosphate hydrolysis [34]. Studies have reported somatic mutations and heterogeneity of the *MYH11* gene in gastric and colorectal tumours [34]. In the current analysis, we also observed differences between STAD and CRC patients when using *MYH11* protein as a prognostic marker, with higher levels of *MYH11* in STAD tumours reflecting significantly poorer survival (Figure 6B) and a relatively better survival rate in CRC tumours (Figure 6C). These results further confirmed the heterogeneity between upper and lower GI cancers. Despite this, proteins of *HER3* and *PR* were identified to be negative prognostic biomarkers in both STAD and CRC patients (Figure 6B,C), which may warrant further studies since they offer significant potential as candidate biomarkers for precision medicine approaches of GI cancers. Elevated *SMAD1* expression was detected in GC tissue and cells; studies demonstrated that *SMAD1* can interact with Yes1-associated transcriptional regulator (*YAP1*) to enhance the cisplatin resistance of GC cells [35]. The adenosine deaminase acting on RNA (*ADAR*) enzymes was associated with the highly aggressive biologic behaviour and poor prognosis in many cancers [36]. Studies indicated that *ADAR* mRNA was elevated and involved in the immune regulator, thus was a novel immune treatment target in CRC [36]. The protein of progesterone receptor (*PR*) is encoded by the progesterone receptor gene (*PGR*), which can modulate the immune response in different cancers [37]. *PGR* expression was reported to be correlated with prognosis and immune cell infiltration in GC [37]. Taken together, the research above reflect that several candidate protein markers may function as potential progression/prognostic biomarkers in GI cancers.

It is not surprising that advanced-stage tumours are associated with worse overall survival [38,39]. In the current study, the results from Cox regression analysis revealed that, in addition to tumour stage, higher age also reflected poor survival in STAD and CRC patients (HR > 1, *p* < 0.05), while a relatively lower BMI value was associated with a better survival rate in CRC patients (HR > 1, *p* < 0.05, Figure 6). Consistent with other studies, ageing was a negative prognostic factor of survival outcome in solid cancer patients [40]. In studies of the association between obesity and survival outcomes in cancer patients, the results from large-scale participants indicated that obesity was associated with more mortality overall [41]. Thus, proper weight loss may represent an effective measure for reducing mortality in cancer patients.

There are some limitations to this study. Firstly, our research was based on retrospective biological data from public databases, and more prospective novel data are necessary to confirm the results, especially to explore the mechanisms and verify the clinical applications of candidate proteins in tumorigenesis and progression. Specifically, we acknowledge that the high percentage of censored cases in CRC (about 75% still alive) may impact the results of our survival analysis. Moreover, the sample size is relatively small and more research with a larger cohort is needed in future studies, to make the results more rigorous. Besides, rather than studying individual proteins, we should and will focus more attention on protein-protein interactions (PPI) that are involved in cancer development.

## 5. Conclusions

In summary, our study provided an overview of the functional proteomic profiles of four major types of GI tract cancer, including ESCA, STAD, COAD, and READ. The similarity in the proteome signature between the two types of lower GI tract cancer, COAD and READ, prompts us to merge them into CRC in follow-up studies. We highlighted the application of feature selection methods during the analysis of high-dimensional biological datasets and further identified several candidate proteins that were correlated with tumour progression and prognosis in ESCA, STAD, and CRC patients. The underlying mechanisms of candidate proteins in tumour development remain poorly understood and warrant more investigation in the future.

## Figures and Tables

**Figure 1 biomolecules-13-00701-f001:**
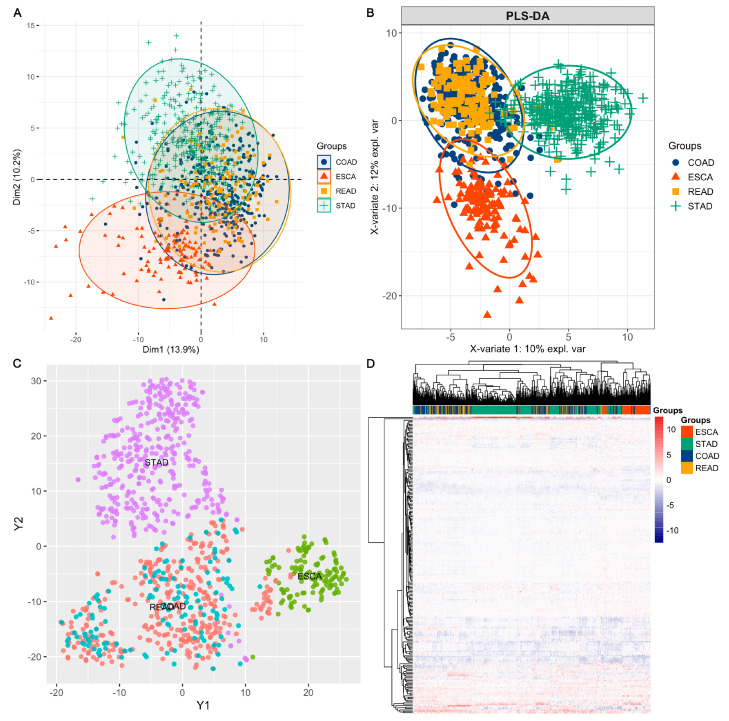
Overview of RPPA functional proteome profiling across ESCA, STAD, COAD, and READ tumour samples: (**A**) PCA plot, (**B**) PLS-DA plot, (**C**) t-SNE, and (**D**) heatmap showing clusters of tumour samples in ESCA, STAD, COAD, and READ based on all 217 protein profiles.

**Figure 2 biomolecules-13-00701-f002:**
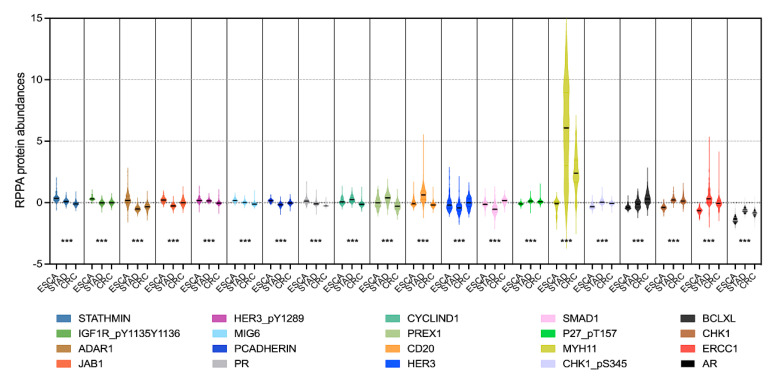
RPPA relative abundances of the 20 selected protein signatures in ESCA, STAD, and CRC samples. Each violin plot shows the minimum, median, and maximum protein abundance in one tumour type. *** *p* < 0.0001 by ANOVA.

**Figure 3 biomolecules-13-00701-f003:**
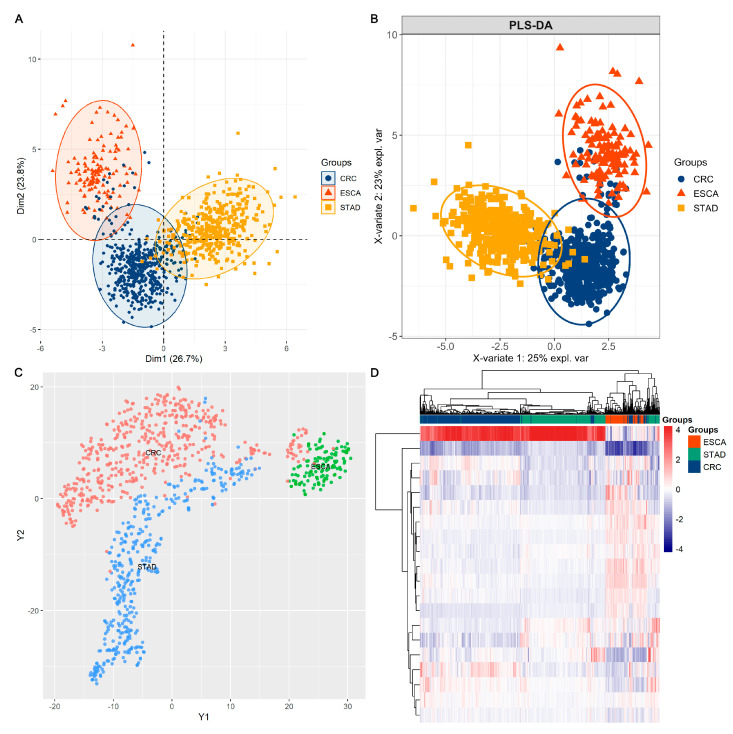
Clusters of tumour samples across ESCA, STAD, and CRC based on the 20 selected protein signatures: (**A**) PCA plot, (**B**) PLS-DA plot, (**C**) t-SNE, and (**D**) heatmap showing clusters of tumour samples in ESCA, STAD, and CRC.

**Figure 4 biomolecules-13-00701-f004:**
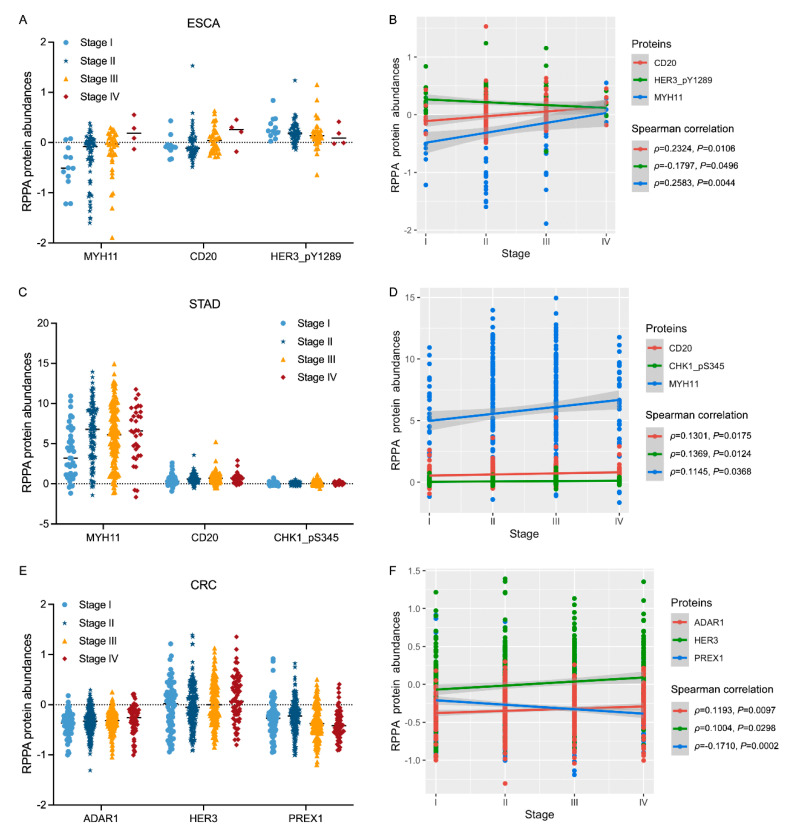
Associations of protein biomarkers with tumour stage: (**A**) scatter plot of proteins *MYH11*, CD20, and *HER_pY1289* in stages I/II/III/IV of ESCA tumours; (**B**) Spearman correlation analysis of proteins *MYH11*, *CD20*, and *HER_pY1289* with tumour stage in ESCA; (**C**) scatter plot of proteins *MYH11*, *CD20*, and *CHK1_pS345* in stages I/II/III/IV of STAD tumours; (**D**) Spearman correlation analysis of proteins *MYH11*, *CD20*, and *CHK1_pS345* with tumour stage in STAD; (**E**) scatter plot of proteins *ADAR1*, *HER3*, and *PREX1* in stages I/II/III/IV of CRC tumours; (**F**) Spearman correlation analysis of proteins *ADAR1*, *HER3*, and *PREX1* with tumour stage in CRC.

**Figure 5 biomolecules-13-00701-f005:**
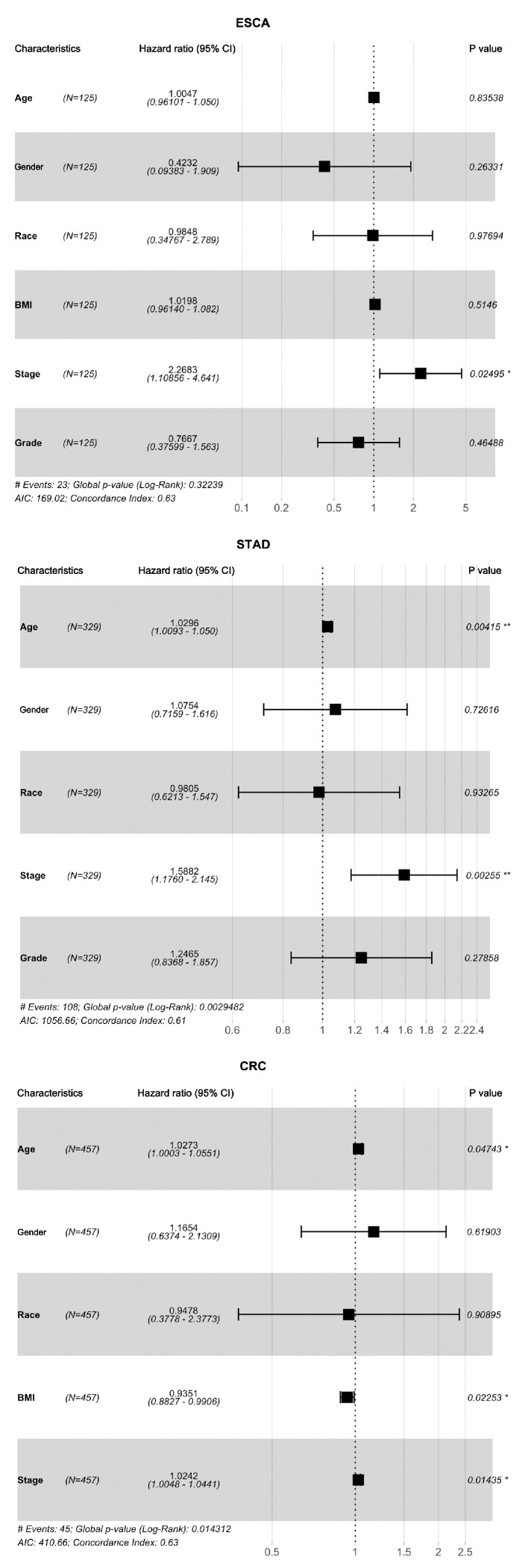
Forest plots of univariate Cox regression analysis in ESCA, STAD, and CRC tumours based on the corresponding clinical parameters. BMI: Body mass index. * *p* < 0.05, ** *p* < 0.01.

**Figure 6 biomolecules-13-00701-f006:**
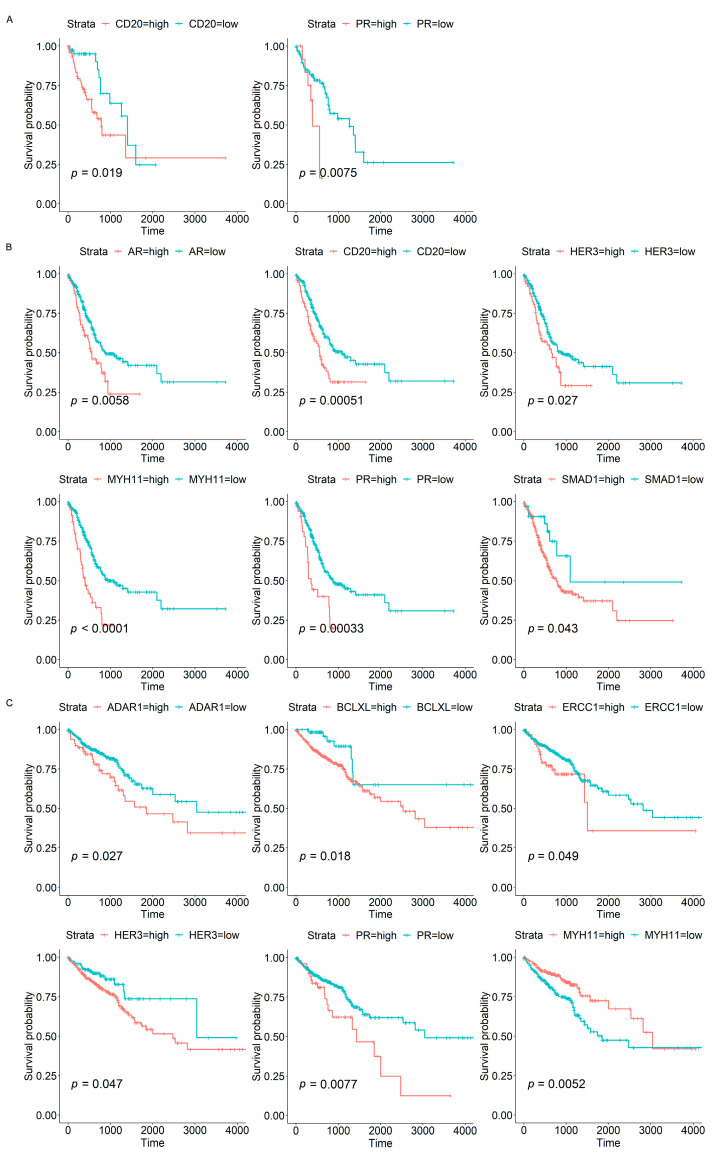
Kaplan-Meier survival curves based on the candidate proteins: Kaplan-Meier survival curves showed that (**A**) two candidate proteins, (**B**) six candidate proteins, and (**C**) six candidate proteins were associated with overall survival in ESCA, STAD, and CRC patients, respectively. The horizontal axis represents survival time in days, and the vertical axis shows the overall survival rate.

**Table 1 biomolecules-13-00701-t001:** Clinical characteristics of ESCA, STAD, and CRC samples in this study.

Clinical Characteristics		Number of Cases		
		ESCA	STAD	CRC
Age at initial pathologic diagnosis (year)	<65	73	151	188
	≥65	53	201	293
	Not reported	0	5	3
Gender demographic	Male	108	236	251
	Female	18	121	230
	Not reported	0	0	3
Race demographic	Asian	43	66	12
	Black	2	5	52
	White	73	232	240
	Not reported	8	54	180
BMI	≤18.4	4	No information	4
	18.5–23.9	60		65
	24.0–27.9	29		74
	≥28	28		117
	Not reported	5		224
Tumour stage	Stage I	12	45	74
	Stage II	66	105	185
	Stage III	38	150	144
	Stage IV	4	33	66
	Not reported	6	24	15
Neoplasm histologic grade	Grade 1	15	9	No information
	Grade 2	55	120	
	Grade 3	32	219	
	Not reported	24	9	
OS_status	Alive	82	189	361
	Dead	43	140	96
	Not reported	1	28	27
OS_time (day)	Alive	576.56 ± 527.05	711.61 ± 573.43	878.75 ± 740.52
	Dead	479.56 ± 427.18	421.28 ± 341.58	677.34 ± 644.99
Total number of cases		126	357	484

BMI: Body mass index; OS: overall survival.

## Data Availability

Not applicable.

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
