# Peer review of "Functional Proteomic Profiling Analysis in Four Major Types of Gastrointestinal Cancers"

_biomolecules, 2023, doi:10.3390/biom13040701_

Round 1

Reviewer 1 Report

This manuscript will greatly benefit from a workflow diagram before the results section that highlights the main/overall process for analyzing the data. This should include for example, the origin of the data (RPPA of patient tumors for example), number of samples, type of samples (disease), how the filtering affected the initial number of samples, then the clustering/bioinformatic methods used and what was obtained from it. This basic diagram will give the reader a main idea before reading in detail the results section with each analysis in detail. 

Reviewer 2 Report

The authors performed statistical analysis on publicly available functional proteomics data from 4 GI cancers.

I have to point out my concerns about methods and there description, figures and conclusions:

1. In the methods section just R packages used are named, however some of those require additional parameters provided by the user, which can significantly alter the output, thus it is hard to interpret the results by the reader.

2. The feature selection section (2.3) on the other hand is too detailed, the first part (line 111-120) would better for introduction or discussion section, not the experimental, the rest with detailed equations is not needed, as those algorithms were not invented or altered by the authors, a publicly avilable R package was applied to the data with published methods.

3. Figure 1. and Figure 3. are quite similar, Fig. 1 just proves the merge of COAD and READ data. But it could be useful to confirm one major statement of the authors in discussion (line 265-283) and conclusion about the usefullnes of feature selection, however no quantitative  measure was given to support that, just subjective visual comparison of those two figures are possible (although parameters used in their creation are missing). There is now explanation why 20 features were selected, how the number of features would effect eg. selectivity.

4.It is also not clear why 3 proteins on Fig. 4., 2-6 proteins on Fig. 6. are shown, what were the filtering criteria.

5. CRC part of Fig. 5. is not visible in the pdf, however if figure would be fitted to page, fonts would be too small to read. So statements on CRC could not be judged. However subtle differences (HR close to1) related to age and BMI (abbreviation may be obvous, but should be defined) are discussed here and in discussion section, but no information is given if results were corrected for general clinical risk factors (age and obesity). 

6. Fonts on Fig. 6. are also too small, protein/gene IDs are hidden and hardly readable.

7. For the survival analysis (Cox and K-M) no statement can be found by the possible bias caused by censored cases (eg. in CRC 75% of cases are still alive).

8. Correlations shown on Figure 4. are really weak (although significant), ESCA stage IV. class includes only 4 cases, so conclusions made on these are not clearly supported.

9. Putative biomarkers selected by the authors are not put into the context of literature data, only two proteins are deatailed, but their possible relation to GI cancers and comparison to previous results of the authors on these cancers are not discussed.

In the current form, I am not convinced about the conlcusions of the authors and I do not see the clear useful novelty.

Round 2

Reviewer 2 Report

Thanks, for the answers and the changes made.